# Maintaining Physicochemical, Microbiological, and Sensory Quality of Pineapple Juice (*Ananas comosus*, Var. 'Queen Victoria') through Mild Heat Treatment

**Charlène Leneveu-Jenvrin** [1,2,*] **, Baptiste Quentin** [1,2]**, Sophie Assemat** [2,3]
**and Fabienne Remize** [1,2]

1    QualiSud, University of La Réunion, CIRAD, Univ Montpellier, Institut Agro, Avignon University,
2 rue J. Wetzell, F-97490 Sainte Clotilde, France; b.quentin2605@gmail.com (B.Q.);
fabienne.remize@univ-reunion.fr (F.R.)

2    QualiSud, Univ Montpellier, Avignon University, CIRAD, Institut Agro, University of La Réunion,
97744 Montpellier, France; sophie.assemat@cirad.fr

3    CIRAD UMR QualiSud, F-97410 Saint Pierre, France

*    Correspondence: charlene.leneveu@univ-reunion.fr

**Abstract:** Shelf life of freshly prepared pineapple juice is short and requires refrigerated conditions of storage. Mild heat treatment remains the easiest way to prolong juice shelf life for small companies. This study was constructed to assess pineapple cv. Queen Victoria juice shelf life from a broad examination of its quality and to propose the most appropriate thermal treatment to increase shelf life without any perceptible decrease in quality. From 25 independent batches of pineapple, collected in different areas and seasons from Reunion Island, the variability of juice physicochemical and microbiological quality was determined. Juice pH values were the highest for fruit harvested in summer, but the juice acidity remained low enough to prevent pathogen spore-forming bacteria growth. During storage at 4 °C, color was modified, and yeasts and molds were the main microbial group exhibiting growth. Assessment of sensory quality resulted in the proposal of a shelf life comprising between three and seven days. Compared to higher temperatures, heat treatment at 60 °C was enough to ensure a good microbiological quality for 30 days, but sensory characteristics and color changes led to the proposal of a shelf life of seven days for pineapple juice treated at 60 °C.

**Keywords:** fruit; beverage; variability; seasonal variations

## 1. Introduction

Pineapple is a tropical fruit crop with several nutritional benefits. It is a good source of vitamin B1, B6, and vitamin C [1,2]. There are many cultivars of pineapple (*Ananas comosus*) in the world, but pineapple 'Queen Victoria' is particularly appreciated for its sweet flavor. Sweetness relies on the ratio between acidity and concentration of sugars [2], but depends also on climatic conditions and cultural practices [3]. Several studies show the impact of climatic conditions on physicochemical parameters of the fruit before harvest, especially in Reunion Island [3,4].

Pineapple is mainly consumed fresh and canned. The commercial 'Queen' cultivar class is not suitable for canning as it results in a large amount of waste due to its morphology. However, it can be processed into juice. Pineapple juice is the third most consumed fruit juice worldwide after orange and apple [5]. Quality and shelf life of untreated juice depend on the raw material and the applied processes [6]. Sensory defects during pineapple juice shelf life result from browning and carotenoid destruction. Similarly, a trained sensory panel described minimally processed pineapple as "sugared", "pineapple", and "fresh" when juice was tasted the day of preparation and "fermented", "alcoholic",

and "chemical" after three to seven days of storage at 4 °C, together with being browner and shinier and losing firmness [7]. Juice is highly susceptible to spoilage, and microbial metabolism leads to deterioration of organoleptic and physicochemical parameters, thereby causing rejection of the product by consumers and financial losses for producers. Browning of juice, resulting from the activity of fruit endogenous enzymes, is the other cause of consumers' rejection [8]. To increase fruit juice shelf life, physical treatments are mostly used as they inactivate both microorganisms and enzymes [9]. Several processing technologies have been applied to fruit juices, such as classical pasteurization, ohmic heating, microwave heating, thermosonication, pulsed electric fields (PEF), light treatment, supercritical carbon dioxide, and high hydrostatic pressure (HHP). For instance, PEF treatment improves shelf life of pineapple juice without compromising nutritional and antioxidant values [10]. These emerging technologies have the potential to preserve the freshness of pineapple juice. However, the equipment required to apply physical treatments still implies a high financial investment, which has to be balanced by innovative advantages of the technology [11]. For many small and medium enterprises, cost remains one of the most significant barriers to innovation meaning they will engage process innovation only if they can be sure it will provide significant financial gains [11,12]. Amongst all technologies, classical pasteurization or mild heat treatments are clearly affordable for small-scale processing units and the ability of these techniques to produce qualitative juices deserves to be more deeply investigated.

The impact of classical mild pasteurization treatment on pineapple juice quality has been only partly described [8,13,14]. In 2014, Hounhouigan et al. [8] concluded on a lack of knowledge of the impact of pasteurization on nutritional and sensory quality of pineapple juice. Pineapple juice pasteurization using hot-fill processing is usually performed at 92–105 °C for 15–30 s. Lower temperature treatments were described such as 75 °C for 3 min, 80 °C for 2 min, or 85 °C for 15 min and allowed storage at ambient temperature [13–15]. These treatments resulted in nonenzymatic browning due to Maillard reaction, pigment destruction, and possibly changes in the content of esters, lactones, furanoids, and carbonyl and sulfur compounds, but barely affected phenolic compounds. On the other side, microbiological quality of unpasteurized and pasteurized pineapple juice has been described more extensively [8,13].

The aim of the present study was to assess the quality of untreated pineapple juice, taking into account batch variability of its characteristics, and then to determine a mild pasteurization treatment so that both the physicochemical characteristics and the sensory quality were preserved through storage at 4 °C.

## 2. Materials and Methods

### 2.1. Sampling

In this study, 25 batches of "Queen Victoria" pineapple were compared. Pineapples were collected from different locations in Reunion Island over two years (2017–2019). For 22 sampling dates and origins, fruits with similar maturity level, corresponding to eyes turning yellow, were picked up (Table 1) [7]. Pineapples were transported in boxes at ambient temperature and stored at temperature 22–25 °C until processing. Three locations were distinguished according to their annual rainfall: west location was characterized by very low rainfall (500–1000 mm), south and north locations by moderate rainfall (1250–2000 mm), and east location by very high rainfall (2000–3000 mm). Whatever the location, average daily solar radiation was high (1700–2100 J/cm$^2$). Two seasons, representative of Reunion Island climate, were observed: winter (colder and dry) from April to September, and summer (warmer and wet) from October to March [7]. In addition, three batches were collected from a local producer of minimally processed fruit, labeled as TP (Table 1) [7]. For the 22 other batches, interpolated meteorological data corresponding to one month before harvest were recovered from CIRAD/Météo-France [16].

**Table 1.** Pineapple batch numbers, sampling location, month of sampling (season), and juice processing. Batches labelled TP were collected from a local producer on minimally processed fruit.

| Batch number | Location | Sampling Month | Juice Processing |
|---|---|---|---|
| 1 | East | October 2017 (summer) | Untreated |
| 2 | East | March 2018 (summer) | Untreated |
| 3 | East | May 2018 (winter) | Untreated |
| 4 | East | June 2018 (winter) | Untreated and Pasteurized |
| 5 | East | June 2018 (winter) | Untreated |
| 6 | East | July 2018 (winter) | Untreated |
| 7 | East | July 2018 (winter) | Untreated |
| 8 | East | July 2018 (winter) | Untreated |
| 9 | West | October 2017 (summer) | Untreated |
| 10 | West | October 2017 (summer) | Untreated |
| 11 | West | March 2018 (summer) | Untreated |
| 12 | West | April 2018 (winter) | Untreated and Pasteurized |
| 13 | West | April 2018 (winter) | Untreated |
| 14 | West | May 2018 (winter) | Untreated |
| 15 | West | May 2018 (winter) | Untreated |
| 16 | West | June 2018 (winter) | Untreated |
| 17 | North | December 2017 (summer) | Untreated and Pasteurized |
| 18 | South | March 2018 (summer) | Untreated |
| 19 | South | March 2018 (summer) | Untreated |
| 20 | South | March 2018 (summer) | Untreated |
| 21 | South | April 2018 (winter) | Untreated |
| 22 | South | May 2018 (winter) | Untreated |
| 23 TP | Any | January 2019 (summer) | Untreated |
| 24 TP | Any | February 2019 (summer) | Untreated and Pasteurized |
| 25 TP | Any | May 2019 (winter) | Untreated and Pasteurized |

### 2.2. Untreated Juice Processing and Sampling

Within 24 h after collection, at least three fruits with similar maturity level were manually peeled and cut, then prepared in juice (Extractor Wismer EW-01, CAPAVENIR, France) and shared into sterile bottles. Each bottle contained 75 mL of juice and was stored at 4 °C until analysis. One bottle was used for each date of analysis.

### 2.3. Pasteurized Juice Processing and Sampling

Within 24 h after collection, at least 17 fruits with similar maturity level were manually peeled and cut, then prepared in juice (Extractor Wismer EW-01, CAPAVENIR, France) so that at least 7 L of juice were obtained. For each batch, 1 L of untreated juice was directly stored at 4 °C (control). Pineapple juice was pasteurized in a Simaco (Bouzonville, France) tubular heat exchanger at 86, 80, 70, or 60 °C for 90 ± 5 s (average residential time) followed by hot fill in 1 L glass bottles previously sterilized. After the bottles were filled and closed with screw caps, they were cooled in a water bath (5 °C), labeled, and then stored under refrigeration (4 °C) until further analysis. One to four replicates were done for each pasteurization temperature. A total of five bottles (for each repetition) were collected for sensory attributes, physical, chemical, and microbiological analyses, before and after the pasteurization, and for each storage time point as indicated.

### 2.4. Microbiological Analyses

From juice samples stored at 4 °C, serial decimal dilutions were performed in SPW (saline peptone water, Condalab, Torrejón de Ardoz, Madrid, Spain). Enterobacteria were enumerated on VRBG agar (Biokar diagnostic, Solabia, Allonne, France) incubated for 48 h at 37 °C. Psychrotrophic bacteria were enumerated on nutrient agar (Merck, Darmstadt, Germany) incubated for three days at 10 °C. Yeasts and molds enumeration was performed on Sabouraud glucose agar with 100 mg/L chloramphenicol

(Biokar diagnostic, Solabia, Allonne, France) after incubation at 30 °C for 5 days. The detection level for these methods was 3 log CFU/mL.

## 2.5. Physicochemical Parameters

A pH meter (5231 Crison, and pH meter Model GLP22, Crison Instruments S.A. Barcelona, Spain) was used to determine pH values. Titration with 0.05 M NaOH (TitroLine easy, Schott, Mainz, Germany) was performed to determine titratable acidity (TA), which was expressed as citric acid equivalents in g/100 mL.

Pineapple juice total soluble solids (TSS), expressed as °Brix, were determined with a hand refractometer (Atago, Tokyo, Japan).

Three color determinations were performed for each sample (12 mL of juice). A spectrophotometer CM 3500d (Minolta®, Carrières-sur-Seine, France) was used to measure the color parameters L*, a*, and b*. Color difference, ΔE, was calculated from numerical values of L*, a*, and b* [7].

## 2.6. Sensory Quality Characteristics

Sensory quality of pineapple juice, placed at room temperature one hour before, was carried out by a panel of trained judges.

For each batch, descriptive profiles were determined from preliminary sessions, which enabled the generation of pineapple descriptive vocabulary [7]. An 11-point scale between 0 (no perception) and 10 (very strong perception) was used by the judges to rate the intensity of the different sensory descriptors. The ISO 11035 method was used [17].

Triangle test was used to determine if there was a detectable difference between two products: control sample and sample pasteurized, both the day of preparation or after seven days of storage according to the ISO 4120–2004 method [18]. Three samples were randomly served and arranged on the plate for evaluation. Each sample was coded differently, with a three-digit code. During this test, the panel was asked to point out their favorite sample.

## 2.7. Statistical Analysis

Statistical treatment of data was performed with the XLSTAT software (Addinsoft, Paris, France).

A *p*-value of 0.001 was used for one-way variance analysis (ANOVA). The REGWQ (Ryan–Einot-Gabriel-Welsh F) test was used for pair-wise comparisons.

Correlation tests were performed with Kendall's tau coefficient with a *p*-value of 0.05 (quantitative variables) or with the biserial correlation method, using the Monte Carlo simulation (correlation between one quantitative and one qualitative variable).

## 3. Results and Discussion

### 3.1. Freshly Prepared Juice Pineapple Characteristics

The 25 pineapple batches were independently transformed into juice. Physicochemical parameters (pH, TA, TSS, L*, a*, and b*, color parameters) and microbial counts were determined on the day of processing for each batch. Table 2 shows mean values and data dispersion between batches.

The juice pH was low, as previously observed for minimally processed pineapple from the same cultivar and location [7]. The maximal pH value was 4.25 and so the juice did not allow *Clostridium botulinum* growth.

TSS values were in a high range, compared to other studies [15,19,20]; for instance, °Brix from 13.1 to 14.4 were described by Sanya et al. [15]. Color parameters showed moderated variability, the widest range corresponding to b* value.

**Table 2.** Physicochemical and microbiological characteristics of untreated pineapple juice on the day of preparation.

| Parameter | Mean | Variation Coefficient | Minimum Value | Maximum Value |
|---|---|---|---|---|
| pH | 3.35 | 8% | 3.08 | 4.25 |
| TA (g/100 mL) | 0.86 | 23% | 0.66 | 1.35 |
| TSS (°Brix) | 15.2 | 9% | 13.0 | 17.6 |
| L* | 69.1 | 9% | 51.0 | 75.0 |
| a* | 2.0 | 70% | −0.5 | 5.6 |
| b* | 50.7 | 17% | 23.6 | 60.1 |
| Psychrotrophic bacteria (log CFU/mL) | 3.3 | 21% | 3.0 [1] | 5.4 |
| Enterobacteria (log CFU/mL) | 3.9 | 26% | 3.0 [1] | 6.5 |
| Yeasts and molds (log CFU/mL) | 4.9 | 8% | 4.0 | 5.5 |

[1]: the indicated values correspond to the detection level. For psychrotrophic bacteria, one batch showed counts below this level. For enterobacteria, five batches showed counts below this level.

Microbiological counts were in the usual ranges for unpasteurized pineapple juice [8]. A large variability between batches, especially for psychrotrophic bacteria and enterobacteria, has to be underlined. The European Union regulation n°2073/2005 requires less than 3 log CFU/mL of *Escherichia coli* for unpasteurized fruit juices. In addition, the levels of yeasts and molds (Y&M) population were high, regarding the usual limit of 6 log (CFU/mL) for fruit juices at consumption level.

The impacts of location and season on batch characteristics were analyzed, taking into account meteorological data (Table 3) [16]. Neither color parameters nor microbiological counts varied according to crop location or harvest season. Meteorological factors (total solar radiation, temperatures, and potential evapotranspiration), juice pH, and TA of pineapple juice were significantly different between north-south, west, and east locations. Juice pH was the highest for fruit harvested in south-north locations, characterized by the highest solar radiation, temperatures, and evapotranspiration. Meteorological factors (total solar radiation, mean temperature, and potential evapotranspiration) and juice pH were significantly different between summer and winter. Biserial correlation showed a correlation between season and pH, with a *p*-value of 0.0001 and a coefficient of 0.70. Summer was characterized by higher solar radiation, temperatures, and evapotranspiration than winter and juice from fruit harvested in summer exhibited higher pH than the juice processed in winter.

**Table 3.** Meteorological data and influence of crop location and harvest season on pineapple juice physicochemical characteristics. For the same column, different letters indicate significant difference.

| | Total Solar Radiation (J/cm$^2$) | Minimal Temperature (°C) | Mean Temperature (°C) | Maximal Temperature (°C) | Potential Evapotranspiration (mm) | pH | TA (%) |
|---|---|---|---|---|---|---|---|
| South-North | 1728 b | 20.0 b | 23.1 b | 28.1 b | 3.69 b | 3.7 b | 0.86 a |
| West | 1425 a | 17.3 a | 21.0 a | 27.1 ab | 2.79 a | 3.4 a | 1.08 b |
| East | 1460 a | 17.9 a | 21.2 a | 25.6 a | 2.92 a | 3.3 a | 0.87 a |
| *p*-value (Location) | 0.018 | 0.026 | 0.083 | 0.066 | 0.007 | 0.013 | 0.060 |
| Summer | 1695 b | 19.4 b | 22.8 b | 28.0 b | 3.56 b | 3.7 b | 0.86 a |
| Winter | 1399 a | 17.5 a | 20.9 a | 25.9 a | 2.75 a | 3.3 a | 1.00 a |
| *p*-value (Season) | 0.001 | 0.027 | 0.016 | 0.013 | 0.000 | 0.000 | 0.117 |

Pearson correlation tests showed positive correlations between pH and maximal (Pearson coefficient 0.735; *p*-value < 0.0001), minimal (Pearson coefficient 0.717; *p*-value = 0.0002), and mean (Pearson coefficient 0.734; *p*-value = 0.0001) temperatures, evapotranspiration (Pearson coefficient 0.630; *p*-value = 0.002), and to a lesser extent, solar radiation (Pearson coefficient 0.510; *p*-value 0.015). This observation is in agreement with previous literature [4], which shows a change in the physicochemical parameters (TSS) of a fruit in relation to the climate one month before harvest. However, no correlation between microbiological counts and physicochemical parameters was noticed.

### 3.2. Physicochemical Characteristics of Untreated Pineapple Juice over Refrigerated Shelf Life

Changes in physicochemical parameters were determined over the shelf life of pineapple juice (Table 4). From the 25 prepared independent batches, three were removed at day 7 and nine more at day 14 because of evident bad smell. During storage, pH, TA, and TSS did not significantly change. This indicates that fermentation did not modify juice properties. Contrarily, color modifications were noticed after three days. The a* green/red and the b* blue/yellow components gradually decreased. As a consequence, a color difference with the day 0 juice was detected after three days of storage and it increased after seven days. Hence, color appears as an indicator of untreated juice quality and a variation during storage could be used as spoilage indicator. Browning of untreated pineapple juice within 15 days was previously reported [21].

**Table 4.** Physicochemical parameters of untreated pineapple juice during storage at 4 °C. For the same line, different letters indicate significant difference.

| Days of Storage | 0 | 3 | 7 | 10 | 14 |
|---|---|---|---|---|---|
| Number of Batches | 25 | 25 | 22 | 22 | 13 |
| **Mean (Minimum Value; Maximum Value)** | | | | | |
| pH | 3.45 (3.08; 4.25) a | 3.40 (3.09; 4.00) a | 3.39 (3.10; 3.96) a | 3.38 (3.12; 3.88) a | 3.24 (3.13; 3.25) a |
| TA (g/100 mL) | 0.93 (0.66; 1.35) a | 0.97 (0.74; 1.25) a | 0.99 (0.67; 1.27) a | 0.99 (0.68; 1.20) a | 1.01 (0.72; 1.26) a |
| TSS (°Brix) | 15.5 (13.0; 17.6) a | 15.6 (12.4; 18.8) a | 15.4 (12.4; 17.6) a | 15.0 (8.2; 17.6) a | 14.9 (12.3; 17.0) a |
| L* | 66.5 (51.0; 75.0) a | 68.5 (56.8;74.0) a | 69.9 (65.1; 74.6) a | 69.2 (63.5; 74.2) a | 69.9 (63.3; 75.2) a |
| a* | 2.3 (−0.5; 5.7) c | 1.4 (−1.2; 3.8) b | 0.8 (−1.3; 2.6) ab | 0.4 (−1.4; 2.2) a | 0.3 (−0.2; 1.0) a |
| b* | 48.1 (23.6; 60.1) b | 45.9 (24.5; 57.7) ab | 42.8 (27.7; 59.0) ab | 41.2 (25.6; 56.2) ab | 41.6 (28.6; 50.4) a |
| Color difference | [1] a | 6.0 (0.7; 22.4) b | 11.1 (1.9; 31.5) bc | 13.2 (2.6; 36.1) c | 11.5 (5.3; 27.8) bc |

[1]: control condition.

### 3.3. Increase of Microbiological Counts of Untreated Pineapple Juice over Refrigerated Shelf Life

The populations of psychrotrophic bacteria, enterobacteria, and Y&M were determined during pineapple juice storage at 4 °C. The average counts of psychrotrophic bacteria did not increase but a gradual increase in maximal population was noticed (Figure 1a). Enterobacteria population was significantly higher than the control (day 0) condition only after 14 days of storage (Figure 1b). Y&M population increase was significant after 10 days, but an increase in population range was noticed from three days of storage (Figure 1c). After three days of storage, one batch reached 6 log (CFU/mL) for Y&M. These observations strengthen the likely involvement of Y&M in untreated juice spoilage, as previously documented for minimally processed pineapple [7].

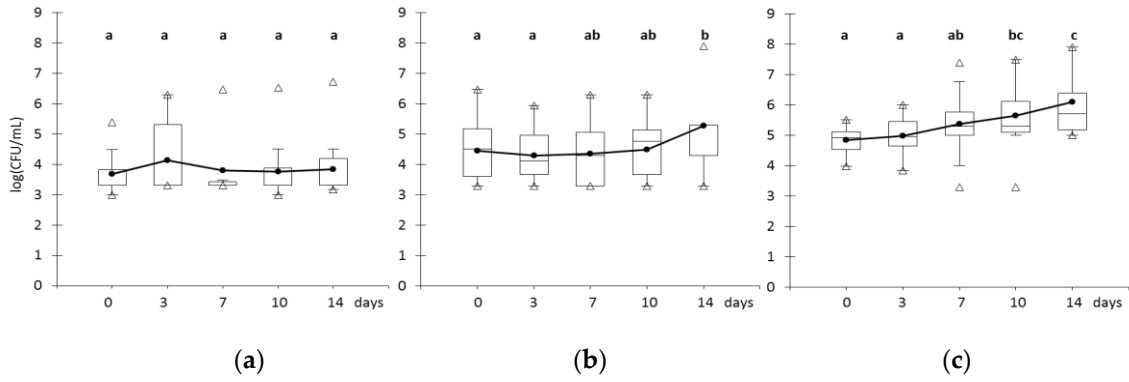

**Figure 1.** Microbial population (log CFU/g) modifications during refrigerated storage of untreated pineapple juice; (**a**) psychrotrophic bacteria, (**b**) enterobacteria, and (**c**) yeasts and molds. Black circles: mean; open triangle: extreme values; boxes: 1st and 3rd quartiles.

In pineapple juice and other pineapple products, the yeasts *Hanseniaspora uvarum* and *Pichia guilliermondii* were the most frequently detected species [22,23]. *Escherichia coli* cannot grow but can

survive several days in pineapple juice [24] and the presence of *Staphylococcus aureus* has been reported in untreated juice [8]. Both *Salmonella enterica* and *Listeria monocytogenes* exhibited population decline in pineapple juice within two days [25]. Considering pineapple juice pH and its variability, the main role of yeasts in spoilage is not surprising. The primary objective of thermal treatment was then to decrease Y&M population levels, together with inactivation of vegetative foodborne pathogens such as *E. coli*.

### 3.4. Sensory Quality of Untreated Pineapple Juice during Refrigerated Storage

A sensory analysis was performed to describe untreated pineapple juice freshly prepared or stored for 3, 7, 10, or 14 days at 4 °C. Olfactive, color, or aspect descriptors with the highest scores are presented in Table 5. For freshly extracted juice or juice stored for three or seven days, the first olfactive descriptors were "Pineapple" and "Sugared", and sensory scores decreased when time of storage increased. For juices stored for 10 or 14 days, descriptors referred primarily to "Fermented" or "Acid".

The overall sensory score of untreated juice the day of preparation and stored for three days at 4 °C was above average, 5.8 and 5.3 (in 10), respectively, followed by juice stored for seven days at 4 °C with an average score of 4.3. From all collected data, a shelf life of untreated juice of three to seven days can be assessed. This choice first results from sensory analysis, strengthened by microbiological data and color modifications.

**Table 5.** Main olfactive, color, or aspect descriptors of freshly prepared and stored untreated juice, in descending order of occurrence. Descriptors used by at least 1/3 of the panel are presented. Mean score values and standard mean errors are added in parentheses.

| Days of Storage | 0 | 3 | 7 | 10 | 14 |
|---|---|---|---|---|---|
| Olfactive descriptors | Pineapple (7.1 ± 0.6) Sugared (5.8 ± 0.8) Fresh (4.8 ± 0.7) Sweet (4.3 ± 0.9) Acid (3.8 ± 0.6) | Pineapple (6.9 ± 0.4) Sugared (5.7 ± 0.7) Fresh (4.2 ± 0.7) Sweet (3.8 ± 0.8) Acid (4.3 ± 0.6) | Pineapple (4.8 ± 0.5) Sugared (5.5 ± 0.7) | Pineapple (3.6 ± 0.7) Sugared (3.4 ± 0.7) Fermented (4.2 ± 0.9) | Pineapple (4.1 ± 0.5) Sugared (4.1 ± 0.7) Fermented (3.3 ± 0.8) |
|  |  |  | Acid (3.8 ± 0.5) | Acid (3.3 ± 0.6) | Acid (4.6 ± 0.7) |
| Color descriptors | Yellow (5.4 ± 0.8) Opaque (4.7 ± 0.7) Dark (4.4 ± 0.5) | Yellow (5.3 ± 0.8) Opaque (4.8 ± 0.7) Dark (4.7 ± 0.5) | Yellow (5.1 ± 0.8) Opaque (4.8 ± 0.6) Dark (4.7 ± 0.6) | Yellow (4.9 ± 0.8) Opaque (5.8 ± 0.6) Dark (6.0 ± 0.3) | Yellow (5.6 ± 0.8) Opaque (5.3 ± 0.6) Dark (5.5 ± 0.3) |
| Aspect descriptors | Dense (3.6 ± 0.7) | Dense (3.5 ± 0.7) | Dense (3.9 ± 0.8) | Dense (4.5 ± 0.7) Lumpy (4.3 ± 0.7) | Dense (4.1 ± 0.7) |

### 3.5. Impact of Mild Heat Treatment on Pineapple Juice Quality

The examination of physicochemical parameters showed that pH, TA, and TSS were not affected significantly by the mild heat thermal processing, and neither by storage time (Supplementary Tables S1–S4). However, color parameters were modified (Figure 2a–c), and the color difference between untreated and pasteurized juice on day 0 was dependent on the pasteurization temperature (Figure 2d).

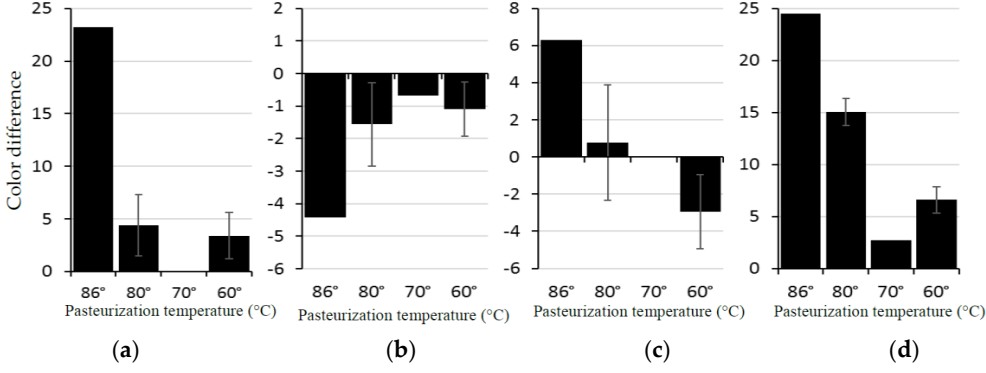

**Figure 2.** Variation of color parameters for different pasteurization temperatures. (**a**) L*, (**b**) a*, (**c**) b* and (**d**) color difference. Each bar represents the difference with the unpasteurized juice. For 86 °C, *n* = 1; for 80 °C, *n* = 2; for 70 °C, *n* = 1; and for 60 °C, *n* = 4.

Browning of pineapple juice caused by thermal treatment has been described [13]. Above 50 °C, nonenzymatic browning and degradation of pigments occur and result in decrease of L* and b* and increase of a*. However, a modification of color parameters of less than 5% was observed for temperatures up to 75 °C, though significant browning was reported after mild thermal treatment (65 °C, 15 min) [21].

During storage, color of the heat-treated juices moderately evolved (Table 6), suggesting that oxidases involved in browning were partly inactivated, even at 60 °C. This observation is not consistent with previous studies of polyphenol oxidase and peroxidase of pineapple showing a 10 to 40% inactivation of enzymes with a treatment of 5 min at 60 °C [26,27]. However, the hot filling process applied here probably resulted in keeping an elevated temperature of juice for an extended time.

**Table 6.** Microbiological counts for unpasteurized and pasteurized pineapple juice at different temperatures and at different refrigerated storage times.

| Processing | Number of Batches | Psychrotrophic Bacteria | | | Enterobacteria | | | Yeasts and Molds | | | Color Difference | | |
|---|---|---|---|---|---|---|---|---|---|---|---|---|---|
| | | Day 0 | Day 14 | Day 30 | Day 0 | Day 14 | Day 30 | Day 0 | Day 14 | Day 30 | Day 0 | Day 14 | Day 30 |
| Control | 1 | 3.3 | 3.3 | − [2] | <3.0 | 5.3 | − [2] | 5 | 7.6 | − [2] | 0 | 19.2 | - |
| Heating 86 °C | 1 | <3.0 | <3.0 | <3.0 | <3.0 | <3.0 | <3.0 | <3.0 | <3.0 | <3.0 | 24.5 | 23.9 | 22.6 |
| Control | 2 | 3.3 | 3.3 [1] | − [2] | <3.0 | 5.3 [1] | − [2] | 5 | 7.6 [1] | − [2] | 0 | 11.5 | - |
| Heating 80 °C | 2 | <3.0 | <3.0 | <3.0 | <3.0 | <3.0 | <3.0 | <3.0 | <3.0 | <3.0 | 15.1 | 15.8 | 17.7 |
| Control | 1 | 5.4 | 6.7 | − [2] | 4.8 | 7.4 | − [2] | 5 | 5.6 | − [2] | 0 | 11.8 | - |
| Heating 70 °C | 1 | <3.0 | <3.0 | <3.0 | <3.0 | <3.0 | <3.0 | <3.0 | <3.0 | <3.0 | 2.7 | 2.4 | 5 |
| Control | 4 | 3.8 | 6.7 [1] | − [2] | 4.8 | 7.4 [1] | − [2] | 5 | 5.6 [1] | − [2] | 0 | - | - |
| Heating 60 °C | 4 | <3.0 | <3.0 | <3.0 | <3.0 | <3.0 | <3.0 | <3.0 | <3.0 | <3.0 | 6.6 | 9 | 8.6 |

[1] for these values, *n* = 1 because of organoleptic spoilage after less than 14 days; [2]: not determined.

As described above, the control conditions (unpasteurized juice) led to microbial population increase. Any temperature assayed resulted in the inactivation of psychrotrophic bacteria, enterobacteria, and Y&M to levels which allowed the absence of detection of these microbial groups after 30 days of storage (Table 6).

Considering the effect of heat treatment on color, which induces browning at 86 °C and at 80 °C (Figure 2), and the absence of detectable microbial growth after 30 days of storage whatever the pasteurization temperature (Table 6), the treatment at 60 °C appears to be the most interesting to preserve juice quality and to increase shelf life.

Heat resistance of yeasts corresponds to D 60 °C of less than 1 min, but *Saccharomyces cerevisiae* presents higher resistance with D 60 °C values ranging between 2.8 and 22 min [8,28]. Inactivation of *E. coli* in pineapple juice is in the range 2.9–5.3 log CFU/mL for a treatment of 5 min at 55 °C [29]. Hence, the treatment applied in this study at 60 °C was enough to decrease these microbial group populations below levels of concern. Other bacteria involved in juice spoilage are spore-forming acidophilic bacteria, such as *Alicyclobacillus acidoterrestris*. However, this bacterium is involved in spoilage of shelf-stable acid beverages. In this study, its growth can be controlled by the low temperature of storage [30,31].

A sensory analysis was performed to compare untreated pineapple juice and 60 °C pasteurized juices. Triangle tests revealed the absence of significant differences between freshly prepared and pasteurized juice (Figure 3a).

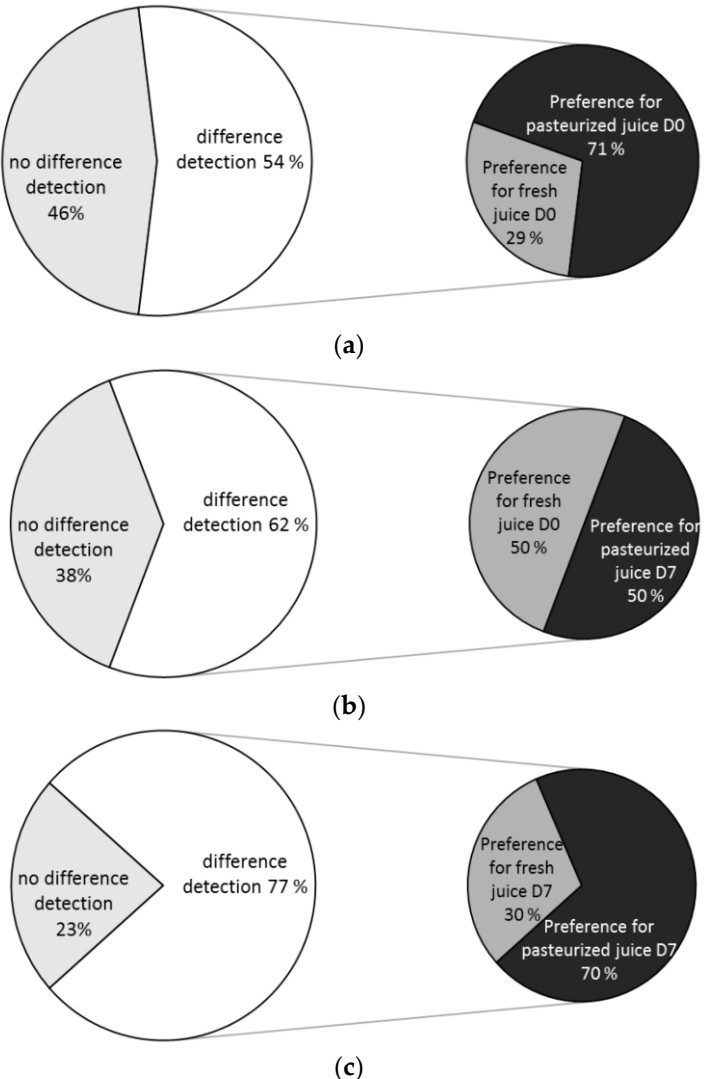

**Figure 3.** Proportion of panelist answers in triangle test comparison of (**a**) untreated juice at day 0 vs. pasteurized juice at day 0, (**b**) untreated juice at day 0 vs. pasteurized juice after 7 days of storage at 4 °C, (**c**) untreated juice after 7 days of storage at 4 °C vs. pasteurized juice after 7 days of storage at 4 °C.

Comparison between untreated juice stored for zero (Figure 3b) or seven (Figure 3c) days with pasteurized juice stored for seven days at 4 °C shows a significant difference at 95% and 99% levels, respectively. For the panelists which detected a difference, there was no preference between untreated juice the day of preparation and seven-day pasteurized juice, but this latter juice was preferred over the seven-day untreated juice.

Organoleptic quality of pineapple juice is rarely reported [8], and the effect of pasteurization on sensory characteristics has not been studied before. We show that a mild heat treatment can be imperceptible by a trained sensory panel, though it is satisfactory for microbiological quality.

## 4. Conclusions

A large variability of pH, TA, TSS, color, and microbial parameters was observed between pineapple juice batches which were sampled from different seasons and locations. Juice acidity was the lowest for fruit harvested in summer, which corresponded to higher mean temperatures, higher solar radiation, and higher potential evapotranspiration than winter. During storage, color parameters and microbial populations changed considerably and affected juice quality. Sensory analyses showed a decrease of quality between three and seven days of storage. A mild thermal treatment at 60 °C was the

best to maintain organoleptic properties close to an untreated pineapple juice. This treatment ensured a good microbiological quality after 30 days at 4 °C, whereas color difference with the untreated juice resulted mainly from the heat treatment but not from post-pasteurization storage change. Sensory assays revealed that the sensory quality of pasteurized pineapple juice was of the same level after seven days of storage as untreated juice.

**Supplementary Materials:** The following are available online at http://www.mdpi.com/2227-9717/8/9/1186/s1, Table S1: Mean values of quality parameters of pineapple juice before and after pasteurization at 60 °C, Table S2: Mean values of quality parameters of pineapple juice before and after pasteurization at 70 °C, Table S3: Mean values of quality parameters of pineapple juice before and after pasteurization at 80 °C, Table S4: Mean values of quality parameters of pineapple juice before and after pasteurization at 86 °C.

**Author Contributions:** Conceptualization and methodology, F.R., C.L.-J., and S.A.; experimental work, C.L.-J. and B.Q.; writing-original draft preparation, C.L.-J. and F.R.; writing—review and editing, F.R., C.L.-J., B.Q., and S.A.; funding acquisition, F.R. All authors have read and agreed to the published version of the manuscript.

**Funding:** This research was funded by European Union and Region Reunion (FEDER), grant number GURDTI-2017-0391-0002361. The APC was funded by the same.

**Acknowledgments:** The authors want to thank Colipays, France and SCA Les Avirons, Vivea, France for providing pineapples.

**Conflicts of Interest:** The authors declare no conflict of interest.

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
