# Peer review of "Maintaining Physicochemical, Microbiological, and Sensory Quality of Pineapple Juice (Ananas comosus, Var. ‘Queen Victoria’) through Mild Heat Treatment"

_processes, doi:10.3390/pr8091186_

Round 1

Reviewer 1 Report

The manuscript is interesting but needs improvement:

Table 3 (the last column) - what is AT?

Line 253- 254: “During storage, color of the heat-treated juices barely changed (Table 6), suggesting that 253 oxidases involved in browning were inactivated, even at 60°C” - This description is inaccurate because as can be seen in Table 6, the color difference increases from 15.8 to 17.7; from 2.7 to 5.0 and from 6.6 to 9.0 for pasteurized samples at 80, 70 and 60 °C, respectively.

Line 265 “Considering the effect of heat treatment on color, which induces browning at 86°C and at 80°C”  -please refer to the table or graph where these results are posted.

In the summary, please indicate what are the conclusions regarding the impact of meteorological data on the selection of appropriate fruit for the production of pineapple juice.

In the supplementary files - no statistical analysis in Tables S1B, S1C and S1D - please correct.

Author Response

We are grateful to the reviewers for their valuable comments. Hereby, the answers are written in red. The modifications in the manuscript appear in revision mode.

Reviewer 1

The manuscript is interesting but needs improvement:

Table 3 (the last column) - what is AT?

This was modified (Titratable Acidity).

Line 253- 254: “During storage, color of the heat-treated juices barely changed (Table 6), suggesting that 253 oxidases involved in browning were inactivated, even at 60°C” - This description is inaccurate because as can be seen in Table 6, the color difference increases from 15.8 to 17.7; from 2.7 to 5.0 and from 6.6 to 9.0 for pasteurized samples at 80, 70 and 60 °C, respectively.

The sentence was modified.

Line 265 “Considering the effect of heat treatment on color, which induces browning at 86°C and at 80°C”  -please refer to the table or graph where these results are posted.

The sentence was modified.

In the summary, please indicate what are the conclusions regarding the impact of meteorological data on the selection of appropriate fruit for the production of pineapple juice.

A sentence was added in the abstract.

In the supplementary files - no statistical analysis in Tables S1B, S1C and S1D - please correct.

As presented in Suppl. Tables B and D, a single batch was assayed, hence performing statistical analyses with only technical errors and not batch variable variability would confuse the reader. Statistical analysis was added in Table C (80°C).

Reviewer 2 Report

Dear editor,

I have carefully reviewed this ms, that appears really interesting and with a complete characterization. In my opinion this paper can be published after the revision of the following minor issues:

  • lines 42-44, the sentence have to be clarified. I suppose the authors are talking about the panel test results. This have to be highlighted;
  • lines 189-90, why did the authors not also analyze the bad smelling samples? I think that a fully characterization of these samples might have been interesting;
  • Table S1, part B and D, please provide the standard deviations . 

Author Response

We are grateful to the reviewers for their valuable comments. Hereby, the answers are written in red. The modifications in the manuscript appear in revision mode.

Reviewer 2

Dear editor,

I have carefully reviewed this ms, that appears really interesting and with a complete characterization. In my opinion this paper can be published after the revision of the following minor issues:

lines 42-44, the sentence have to be clarified. I suppose the authors are talking about the panel test results. This have to be highlighted;

The sentence was modified.

lines 189-90, why did the authors not also analyze the bad smelling samples? I think that a fully characterization of these samples might have been interesting.

We thank the reviewer for this relevant remark. We decided not to propose the obviously spoiled samples to the panel as the juice odor was really strong and might have introduce a bias is the comparison of consumable samples.

Table S1, part B and D, please provide the standard deviations.

As presented in Suppl. Tables B and D, a single batch was assayed, hence performing statistical analyses with only technical errors and not batch variable variability would confuse the reader. Statistical analysis was added in Table C (80°C).

Reviewer 3 Report

The manuscript describes the quality assessment of untreated pineapple juice and the effects of mild pasteurization treatment. The experimental approach is based on determinations of pH, titrable acidity, total soluble solids, color parameters, sensory analysis and several microbiological analyses. Overall, there are no new findings, but some of the obtained results can be of interest for processors.

Suggestions for an improved version:

L.76 – replace „Fruit collection “ > Sampling

In 2.1 – missing relevant details about transportation and storage from the sampling site to the laboratory – consider adding them

L.89 – replace ”Date” > Sampling month

L.93, 113, 118, 121- replace all instances of ‘’ml’’ > mL

L.130 - move ’’ ISO 11035 (Sensory analysis - Identification and selection of descriptors for establishing a sensory profile by a multidimensional approach)’’ to References and cite this in a proper way

L 135 – move the issue related with ISO 4120... to References and cite this in a proper way

L.179 – missing values of Pearson’s correlation coefficients – consider adding them

L.224 – only a subjective and partial sensory analysis was accomplished; despite the authors mentioned they used an eleven levels’ scale, scores were not mentioned for descriptors; the scores from lines 232-234 have no data support – consider adding relevant information

L.293 – missing data for the conclusion ”The analysis of the meteorological data (location and season) on batches characteristics revealed a correlation between season and pH values” – consider adding them or remove this conclusion

Overall, English needs improvement, as well as technical language, in several instances

Author Response

We are grateful to the reviewers for their valuable comments. Hereby, the answers are written in red. The modifications in the manuscript appear in revision mode.

Reviewer 3

The manuscript describes the quality assessment of untreated pineapple juice and the effects of mild pasteurization treatment. The experimental approach is based on determinations of pH, titrable acidity, total soluble solids, color parameters, sensory analysis and several microbiological analyses. Overall, there are no new findings, but some of the obtained results can be of interest for processors.

Suggestions for an improved version:

L.76 – replace „Fruit collection “ > Sampling

The sentence was modified (L. 81 in the modified version).

In 2.1 – missing relevant details about transportation and storage from the sampling site to the laboratory – consider adding them

A sentence was added L. 90-91.

L.89 – replace ”Date” > Sampling month

This was modified in Table 1.

L.93, 113, 118, 121- replace all instances of ‘’ml’’ > mL

This was modified.

L.130 - move ’’ ISO 11035 (Sensory analysis - Identification and selection of descriptors for establishing a sensory profile by a multidimensional approach)’’ to References and cite this in a proper way

Citation was added.

L 135 – move the issue related with ISO 4120... to References and cite this in a proper way

Citation was added.

L.179 – missing values of Pearson’s correlation coefficients – consider adding them

Pearson coefficients were added.

L.224 – only a subjective and partial sensory analysis was accomplished; despite the authors mentioned they used an eleven levels’ scale, scores were not mentioned for descriptors; the scores from lines 232-234 have no data support – consider adding relevant information

The scale was specified in the section Materials and methods (L. 135-136) and data were added in Table 5 (mean values and SME).

Line 232-237 were modified to better reflect table data.

L.293 – missing data for the conclusion: ”The analysis of the meteorological data (location and season) on batches characteristics revealed a correlation between season and pH values” – consider adding them or remove this conclusion

The result section L. 177-183 was clarified, and the conclusion was modified accordingly.

Overall, English needs improvement, as well as technical language, in several instances

We modified the text to improve English.